# The motivation for physical activity is a predictor of VO$_{2peak}$ and is a useful parameter when determining the need for cardiac rehabilitation in an elderly cardiac population

**Nicolai Mikkelsen**📷*, **Christian Have Dall**📷**, Marianne Frederiksen, Annette Holdgaard, Hanne Rasmusen, Eva Prescott**

Bispebjerg Frederiksberg Hospital, University of Copenhagen, Copenhagen, Denmark

* nicolaimikkelsen@hotmail.com

## Abstract

**Data Availability Statement:** Raw data is located in a locked server in Frederiksberg Hospital and

### Background

Exercise-based cardiac rehabilitation (CR) is an essential contributor to a successful recovery for elderly cardiac patients. The motivation for physical activity is a psychological parameter seldom described in secondary prevention, and it is plausible that motivation contributes to the differential effect of CR.

### Purpose

To investigate if motivation, measured using the behavioural regulation in an exercise questionnaire (BREQ-2), predicts VO$_{2peak}$ in elderly cardiac patients before and after CR.

### Methods

A prospective cohort study of elderly ischemic cardiac patients and patients with valvular disease participating in cardiac rehabilitation was used. Motivation was measured using BREQ-2, which measures five constructs of motivation and a summed score—the relative autonomy index (RAI). VO$_{2peak}$ was measured before and after CR using a cardiopulmonary exercise test (CPET).

### Results

Two hundred and three patients performed the baseline tests and initiated CR. One hundred and eighty-two completed CR and comprised the follow-up group. The mean VO$_{2peak}$ was 18 ml/kg/min (SD±5.1). VO2peak increased significantly with increasing motivation, 1.02 (.41–1.62) ml/kg/min pr. SD. Mean improvement from CR was 2.3 ml/kg/min (SD±4.3), the equivalent of a 12% increase. A change in VO$_{2peak}$ after CR was likewise positively associated with increased motivation, .74 (.31–1.17) pr. SD.

cannot be shared publicly due to patient sensitive data and according to Danish data protection. Contact information for future researchers: Request for access to data: The Danish Health Data Authority Ørestads Boulevard 5 2300 København S Denmark Email: forskerservice@sundhedsdata.dk Telephone: +4540357909 Location of data: Frederiksberg Hospital Department of Cardiology 2000 Frederiksberg Denmark Email: hjerteafdelingen.bbh-frh@regionh.dk Phone number: +4538163003.

**Funding:** The author(s) received no specific funding for this work.

**Competing interests:** The authors have declared that no competing interests exist.

## Conclusion

The level of motivation predicts $VO_{2peak}$ before CR, and is also able to predict changes in $VO_{2peak}$ following CR. Motivation measured with the BREQ-2 questionnaire can be applied as a screening tool for elderly cardiac patients before they initiate CR to identify patients with need of specific attention.

## Introduction

Cardiac rehabilitation (CR) is an essential contributor to a successful recovery for cardiac patients. CR improves life expectancy, physical function, and quality of life and patients experience less relapse of the disease compared to non-participants of CR [1–3] Although the benefits of exercise-based rehabilitation are well studied, compliance and adherence in CR remain low [4]. This is especially the case in elderly cardiac patients [5]. Poor compliance and adherence have previously been linked to psychological distress, especially anxiety and depression [6–8]. As psychological distress is more prevalent in the younger segment of the cardiac population [9, 10], other psychological factors may determine success in CR in the elderly cardiac patients.

Physical exercise is a primary component in CR. The 'gold standard' of measuring effect of the exercise component of CR is VO2peak from a symptom-limited cardiopulmonary exercise test (CPET). VO2peak is a precise measure and a reliable individual predictor of future health outcomes such as CVD and mortality [11–14]. The motivation for physical activity is a parameter seldom described in secondary prevention, and it is plausible that the lack of motivation to be physically active can explain the difference in success rates, assessed as improvement of $VO_{2peak}$, when participating in CR.

The motivation for physical activity can be measured using the validated "behavioral regulation in exercise questionnaire" (BREQ-2). BREQ-2 is based on the Self Determination Theory, which is used to understand exercise and physical activity patterns [15, 16] and why people adopt and/or maintain a behavior change [17, 18]. The BREQ-2 questionnaire measures different constructs of motivation. A summed score of the level of motivation can be derived by combining the constructs of motivation. The summed score is named the Relative Autonomy Index (RAI).

This study aimed to investigate if the constructs of motivation and/or the RAI was associated with the effect of CR, measured as $VO_{2peak}$, in elderly cardiac patients.

## Methods

### The study population

This is a prospective cohort study of elderly cardiac patients entering a CR program at a Danish cardiac rehabilitation unit at a hospital in Copenhagen from December 2015 to February 2018 [19]. Patients were asked to join the study if they were more than 64 years of age and met one of the following criteria within three months of entering the CR program: 1) had acute coronary syndrome, including myocardial infarction 2) underwent percutaneous coronary intervention, 3) received coronary artery bypass grafting or 4) received a heart valve replacement. Exclusion criteria: Patients with a contraindication to CR, mental impairment leading to an inability to cooperate, a severely impaired ability to exercise, signs of severe cardiac ischemia and/or a positive exercise testing on severe cardiac ischemia, insufficient knowledge of the

native language and an implanted cardiac device (CRT-P, ICD). For the majority of the patients this was their first enrolment in CR because patients attending a second CR would be referred to municipal CR.

Ethical approval for the study was obtained from the Regional Scientific Ethical Committee for Copenhagen, Denmark (Ref.: H-15011913) and the study was conducted in accordance with the Declaration of Helsinki. Written informed consent was obtained.

The study cohort is part of a multi-center/national cohort study, *European Cardiac Rehabilitation in the Elderly* (EU-CaRE, [20]).

### Rehabilitation program

The CR program consisted of a supervised eight-week outpatient exercise intervention at a hospital with two weekly sessions (16 sessions in total) of 1.5 hours with a high-intensity interval (80% of $VO_{2peak}$) and resistance training. Patients were instructed in self-monitoring of training intensity using the Borg Scale and the sessions were supervised by an instructor. The training sessions were not routinely monitored with heart rate sensors. The program was complemented with a weekly 1.5 hour session of group-based patient education on cardiovascular disease, psychological issues, and diet counseling. Additionally, patients had one or more individual sessions with a cardiologist, a dietician, a physiotherapist, and a nurse.

### Primary exposure–The motivation for physical activity

The level and type of motivation were measured using BREQ-2 (see S1 Appendix). The BREQ-2 is the second version of the questionnaire and is a validated and useful tool to measure a patient's motivation for exercise [21].

The BREQ-2 inventory comprises 19 items. Each item has five possible answers scored on a scale of 0–4 (0 = Not true for me; 4 = Very true for me). The questionnaire assesses five constructs: *amotivation* –e.g., "I think that exercising is a waste of time"; *external regulation* –e.g., "I exercise because other people tell me I should"; *introjected regulation* –e.g., "I feel guilty when I do not exercise"; *identified regulation* –e.g., "I value the benefits/advantages of exercising"; and *intrinsic motivation* –e.g., "I enjoy my exercise sessions".

BREQ-2 was measured as a multidimensional scale, measuring each of the five types of motivation. Additionally, a summed score was derived from the five subscales, the Relative Autonomy Index (RAI). The RAI gives an index of the degree to which respondents are motivated. The RAI is obtained by weighting each subscale and then summing these weighted scores. Each subscale score is multiplied by its weighting, and then the weighted scores are summed. Higher, positive scores indicate greater relative autonomy; lower, negative scores indicate more controlled regulation.

In the descriptive analyses (Table 1) the RAI was categorized into low, medium and high degree of motivation. Since there is no recommended categorization of RAI, we arbitrarily created cut-points that yielded a reasonable distribution of the population.

### Study outcome

The primary study outcome was $VO_{2peak}$. $VO_{2peak}$ was assessed before and after CR using a cardiopulmonary exercise test (CPET) using a maximal symptom-limited bicycle ergometer test (Via Sprint 150P, Ergoline). Breathing gases were collected and analyzed (Jaeger, Master Screen, vers.5.21, Cardinal Health). Each test aimed at a respiratory exchange ratio greater than 1.1 to ensure the validity of the CPET tests [23]. $VO_{2peak}$ was defined as the highest value of oxygen consumption reached, despite progressive increase of the load applied, with the

**Table 1. Patient characteristics by level of motivation at the baseline.**

| | Total population | Motivation for physical activity | | | p value |
|---|---|---|---|---|---|
| | | Low (RAI<0) | Medium (RAI 0–9) | High (RAI 10–20) | |
| **N** | **203** | **46 (23%)** | **94 (46%)** | **63 (31%)** | |
| **Age (years)** | 72.3±5 | 71.3±5 | 72.2±5 | 72.8±5 | 0.360 |
| **Sex (male)** | 149 (73%) | 34 (74%) | 69 (73%) | 46 (73%) | 0.995 |
| **Body mass index (kg/m$^2$)** | 27.3 (4.6) | 29.3 (4.7) | 26.8 (4.0) | 26.3 (4.4) | ***<0.001*** |
| **Living status (alone)** | 66 (33%) | 21 (46%) | 27 (29%) | 18 (29%) | 0.096 |
| **Ethnicity (Non- Western European)** | 12 (6%) | 2 (4%) | 4 (4%) | 6 (9%) | 0.342 |
| **Educational attainment** | | | | | |
| **Short-term education** | 107 (54%) | 26 (57%) | 43 (46%) | 38 (60%) | 0.169 |
| **Long-term education** | 96 (46%) | 20 (43%) | 51 (54%) | 25 (40%) | |
| **Index event** | | | | | |
| **ACS** | 101 (50%) | 17 (37%) | 48 (52%) | 36 (57%) | 0.200 |
| **Stable CAD** | 73 (36%) | 22 (48%) | 34 (36%) | 17 (27%) | |
| **Heart valve replacement** | 28 (14%) | 7 (15%) | 11 (12%) | 10 (16%) | |
| **Smoking status** | | | | | |
| **Never smoked (>1year)** | 99 (49%) | 27 (59%) | 58 (62%) | 14 (63%) | 0.094 |
| **Former smoker (<1year)** | 60 (30%) | 14 (30%) | 29 (31%) | 17 (27%) | |
| **Smoker** | 18 (21%) | 5 (11%) | 7 (7%) | 6 (10%) | |
| **Hypertension (yes)** | 83 (67%) | 14 (70%) | 32 (66%) | 37 (59%) | 0.468 |
| **Hypercholesterolemia (yes)** | | 35 (76%) | 66 (71%) | 35 (55%) | ***0.047*** |
| **Ejection fraction (%)** | 51% (9.7) | 50% (9.6) | 52% (9.5) | 53% (10.0) | ***0.031*** |
| **Diabetes (yes)** | 39 (20%) | 16 (35%) | 15 (16%) | 8 (13%) | ***0.009*** |
| **Peripheral artery disease (yes)** | 19 (9%) | 4 (9%) | 11 (12%) | 4 (6%) | 0.485 |
| **COPD (Yes)** | 15 (7%) | 2 (4%) | 8 (9%) | 5 (8%) | 0.666 |
| **Kidney disease (yes)** | 24 (12%) | 5 (11%) | 11 (12%) | 8 (13%) | 0.978 |
| **Beta blockers (yes)** | 142 (70%) | 34 (74%) | 64 (68%) | 44 (70%) | 0.779 |
| **Statins (yes)** | 178 (87%) | 40 (87%) | 83 (88%) | 55 (87%) | 0.969 |
| **Vital exhaustion (0–17 score)** | 4.4 (4.0) | 5.6 (4.1) | 4.2 (4.1) | 3.6 (3.4) | ***0.027*** |
| **PHQ-9 (0–27 score)** | 5.2 (4.5) | 5.8 (4.6) | 5.2 (4.8) | 4.4 (4.3) | 0.641 |
| **GAD-7 (0–21 score)** | 3.3 (4.2) | 4.1 (4.4) | 3.4 (4.5) | 2.6 (3.4) | 0.607 |
| **VO$_{2peak}$ (mL/kg/min) before CR** | 18.0 (5.2) | 16.1 (3.9) | 17.5 (4.7) | 20.3 (6.0) | ***<0.001*** |
| **RER** | 1.09 (0.1) | 1.09 (0.1) | 1.07 (0.1) | 1.1 (0.1) | 0.398 |
| **Borg score (0–20)** | 15.5 (3.7) | 14.3 (5.1) | 16.0 (2.6) | 15.1 (3.7) | 0.702 |
| **Predicted VO$_{2peak}$ before CR*** | 15.7 (7.0) | 12.6 (5.6) | 15.8 (5.9) | 18.1 (8.3) | ***<0.001*** |

Abbreviations: RAI, relative autonomy index; ACS, acute coronary syndrome; CAD; coronary artery disease; COPD; chronic obstructive pulmonary disease; PHQ-9; patient health questionnaire; GAD-7; generalized anxiety disorder; RER; respiratory exchange ratio; CR; cardiac rehabilitation. Data are reported as mean ± SD or number (%).

*Predicted VO$_{2peak}$ is derived from the validated prediction model by Myers et al [22]

development of a plateau in the VO2 curve during the CPET. When a plateau was not identified, the highest value obtained at the end of test was characterized as VO$_{2peak}$.

Patients that either withdrew their consent or did not attend the CPET after CR were considered as prematurely ending the program. Compliance was defined according to proportion of planned training- sessions attended (<50%, 50–75% or >75% attendance).

## Confounding variables

Other variables of interest included age, sex, revascularization (PCI or CABG), educational attainment (Short or higher education), working status (working or retired), smoking status (never a smoker, former smoker, current smoker), physical activity level during leisure time (>30 minutes, 0–7 days per week) before cardiac event, the use of beta blockers (yes/no), use of statins (yes/no), left ventricular ejection fraction (%), comorbidities. Psychological distress was accounted for with three questionnaires measuring vital exhaustion, depression, and anxiety. Vital exhaustion was assessed using a 17-item questionnaire [24]. Depression and anxiety were assessed the validated Patient Health Questionnaire (PHQ-9) [25] and General Anxiety Disorder questionnaire (GAD-7) [26], respectively. Information on history of hypertension, hypercholesterolemia and co-morbidities were based on hospital records.

Follow-up analyses were also investigated for the influence of $VO_{2peak}$ at baseline, the premature end of rehabilitation and compliance to the CR program

## Statistical analysis

The summed motivation score, RAI, is a continuous variable. For the descriptive statistics, the RAI was categorized into low (RAI<0, medium (RAI 0–9) and high (RAI>10–20) levels of motivation. For the inferential statistics, the RAI was used as a continuous variable.

Normally distributed variables were compared across the different levels of motivation using one-way ANOVA. Non-normally distributed variables were tested using Mann-Whitney and Kruskal-Wallis tests, while a Chi 2 test tested categorical data. Statistically significant differences between the groups were tested with pairwise comparisons using t test and Chi 2 tests.

Correlation between the constructs of motivation and $VO_{2peak}$ were assessed by scatterplots and tested using Pearson's Correlation.

The influence of motivation on $VO_{2peak}$ was tested using multiple adjusted linear regression analyses. Confounders for both baseline and follow-up analyses were identified according to previous literature [27, 28] and whether they were associated with CRF and motivation. Follow-up analyses were additionally adjusted for $VO_{2peak}$ at baseline and compliance to the CR program. Identified confounders were tested sequentially against a simple regression model and adjusted for sex and age to assess the impact on CRF. Confounders that influenced a change in the estimate for motivation by more than 15% were included in the final model [29]. Due to the different continuous scales of the covariates, a standardized regression model was conducted to compare the strength of the association of different continuous predictors with the outcome within the same model.

A 2-tailed $p$ value <0.05 was considered to be statistically significant. All statistical analyses were carried out using STATA IC 13.1 (StataCorp LP).

# Results

Two hundred and thirty-seven patients were initially included in the study. Two hundred and three performed the baseline CPET and completed the BREQ-2 questionnaire. These patients comprised the baseline analyses. One hundred and eighty-two patients performed the second CPET and comprised the follow-up analyses. For a detailed overview of patient exclusion, see S1 Flowchart of the patient population with the number of patients excluded and the reason for exclusion.

## Baseline characteristics

Baseline characteristics, according to the categorized RAI score (low, medium and high motivation), are presented in Table 1.

The mean age of the population was 72 (±5) years old, and 73% was male. The majority were Western European and living with a spouse. Almost half of the population had a higher educational attainment. Only 9% were current smokers. Half of the population had a PCI, 30% CABG, 14% heart valve replacement and 5% no revascularization. The mean $VO_{2peak}$ before CR was 18.2 (±5.0). The mean motivation RAI score was 5 (±7) ranging from -14 to 19.

Twenty-three percent of the population had low levels of motivation, while 31% were highly motivated. Patients with a low level of motivation had an overall higher burden of risk factors: higher body mass index, higher prevalence of both diabetes and hypertension, and lower LVEF. Patients with a low level of motivation also tend to live alone more and score higher on vital exhaustion than patients with both medium and high levels of motivation.

Differences between high and medium motivation levels were minor: Patients with a high level of motivation had a lower prevalence of hypercholesteremia, less hypertension, and lower vital exhaustion score.

There was a definite increase in $VO_{2peak}$ with an increasing level of motivation (p<0.001).

## Constructs of motivation

A correlation between the five constructs of motivation and $VO_{2peak}$ was tested using pairwise correlations and scatterplots with a linear, see Fig 1.

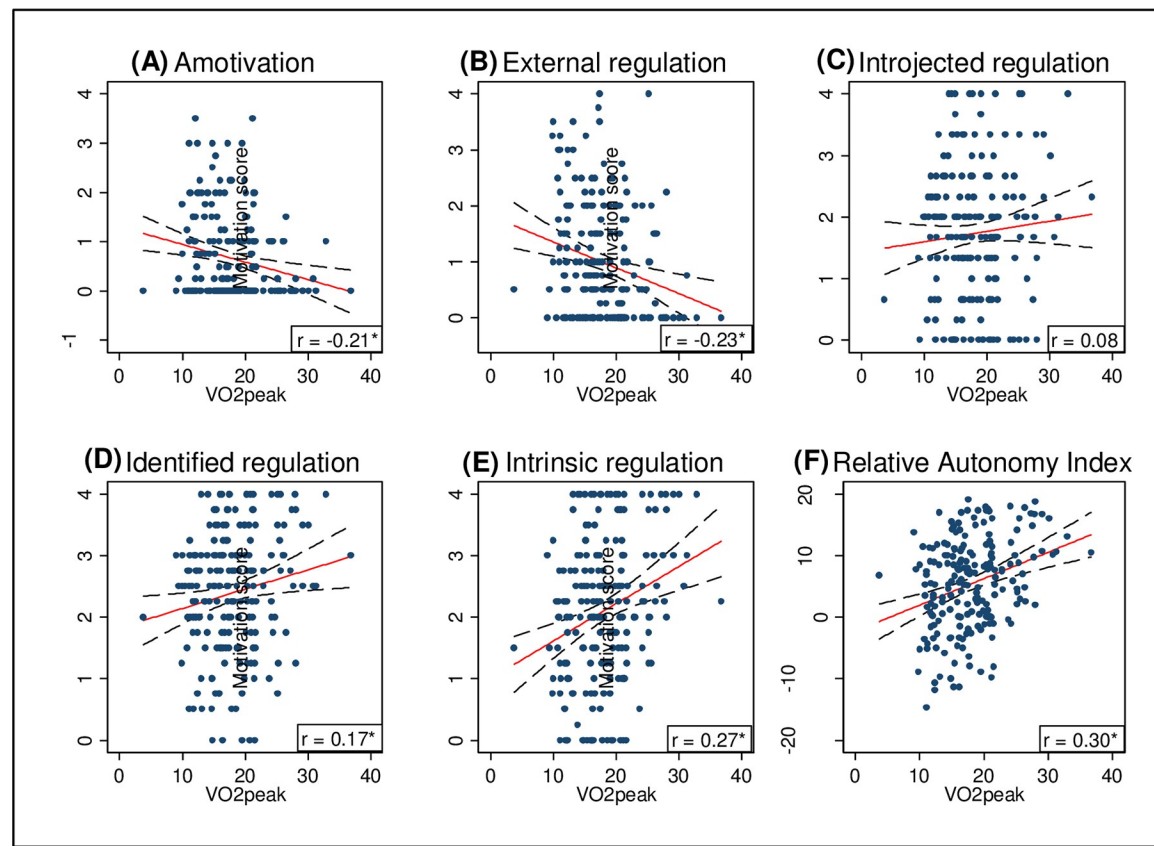

**Fig 1. Correlation between different constructs of motivation and VO2peak with Spearman coefficient (r).** (A) Amotivation: relationship between amotivation score and $VO_{2peak}$. r = -0.21*; (B) External regulation: relationship between external regulation and $VO_{2peak}$. r = 0.23*; (C) Introjected regulation: relationship between introjected regulation and $VO_{2peak}$. r = 0.08; (D) Identified regulation: relationship between identified regulation and $VO_{2peak}$. r = 0.17*; (E) Intrinsic regulation: relationship between intrinsic regulation and $VO_{2peak}$. r = 0.27*; (F) Relative autonomy index: relationship between relative autonomy index and $VO_{2peak}$. r = 0.30*. Significance: *P<0.05.

Except for introjected regulation, all constructs of motivation were significantly correlated with $VO_{2peak}$ at baseline. $VO_{2peak}$ correlated most with RAI (r = 0.30, p< 0.01). RAI also correlated with the other constructs of motivation. The highest correlation was observed between RAI and $VO_{2peak}$. Amotivation and external regulation were especially right-skewed, whereas intrinsic regulation had a bimodal distribution.

To avoid multicollinearity between the different constructs of motivation, the RAI was chosen as the primary exposure variable for the baseline analyses.

## $VO_{2peak}$ before CR

$VO_{2peak}$ was associated with sex, activity level, revascularization procedure, comorbidity, and vital exhaustion, but not with age, ejection fraction, or use of beta-blockers. RAI and $VO_{2peak}$ were significantly correlated (Fig 1, plot (F)). When adjusting for age and sex, the RAI remained significantly associated with $VO_{2peak}$ (1.70 per SD) (Table 2). After multiple

**Table 2. Standardized coefficients of predictors of $VO_{2peak}$ before cardiac rehabilitation (A) and a change in $VO_{2peak}$ following cardiac rehabilitation (B).** Values indicate the difference in ml/kg/min.

| A | Age and sex adjusted | | Multiple adjusted model | |
|---|---|---|---|---|
| Motivation (RAI pr. SD) | 1.70 | (1.04–2.37) *** | 1.05 | (.43–1.69) ** |
| Age (pr. SD) | -1.47 | (-3.09 - .15) | -1.02 | (-2.47 - .42) |
| Sex (male) | 2.65 | (1.06–4.24) *** | 3.23 | (1.85–4.60) *** |
| Activity level (>30 min.) | 0 days | | 0 days | |
| 2–4 days | 1.71 | (-.04–3.45) | 1.01 | (-.52–2.56) |
| 5–7 days | 3.54 | (1.93–5.15) *** | 2.64 | (1.18–4.10) *** |
| Index diagnosis | ACS | | ACS | |
| Stable CAD | -1.76 | (-3.29 - .23) * | -.90 | (-2.25 - .45) |
| Heart valve replacement | -2.47 | (-4.62 - .32) * | -1.80 | (-3.69 - .09) |
| Ejection fraction (pr. SD) | 1.28 | (.46–2.09) * | 1.24 | (.54–1.94) ** |
| COPD (yes) | -2.34 | (-5.03 - .35) | -2.09 | (-4.35 - .17) |
| Diabetes (yes) | -4.01 | (-5.71 - -2.30) *** | -2.22 | (-3.81 - -.64) * |
| Kidney disease (yes) | -3.98 | (-6.14 - -1.82) *** | -3.24 | (-5.13 - -1.35) ** |
| PHQ-9 (pr. SD) | -1.28 | (-1.99 - -.58) *** | -.89 | (-1.51 - -.28) * |
| **B** | **Age and sex adjusted** | | **Multiple adjusted model** | |
| Motivation (RAI pr. SD) | .57 | (.12–1.01) * | .78 | (.33–1.24) ** |
| Baseline $VO_{2peak}$ (pr. SD) | -.56 | (-1.16 - .21) | -1.30 | (-1.94 - -.66) *** |
| Age (pr. SD) | -.84 | (-1.85 - .18) | -1.00 | (-2.01 - -0.01) * |
| Sex (male) | -.18 | (-1.21 - .85) | .74 | (-.28–1.76) |
| Smoking status | Never smoker | | Never smoker | |
| Previous smoker | -.23 | (-1.22 - .77) | -.37 | (-1.34 - .59) |
| Current smoker | -1.43 | (-3.00 - .14) | -1.89 | (-3.36 - -.40) * |
| Diabetes (yes) | -1.37 | (-2.50 - -.24) * | -1.62 | (-2.78 - -.48) ** |
| Kidney disease (yes) | -0.98 | (-2.37–0.42) | -1.45 | (-2.84 - -0.06) * |
| PHQ-9 (pr. SD) | .22 | (-.24 - .69) | .21 | (-.24 - .66) |

Significance levels

*p<0.05

**p<0.01

***p<0.001.

Abbreviations: RAI, relative autonomy index; ACS, acute coronary syndrome; CAD; coronary artery disease; COPD; chronic obstructive pulmonary disease; PHQ-9; patient health questionnaire.

adjustments, motivation remained associated with $VO_{2peak}$ (1.05 per SD), even after an adjustment for depression. Being male was also positively associated (3.23ml/kg/min) as was being physically active 5–7 days per week (2.64 ml/kg/min) compared to 0 days per week. Chronic kidney disease was associated with a lower baseline VO2peak (-3.34 ml/kg/min) as were diabetes and COPD (-2.22 and -2.09 ml/kg/min, respectively).

Motivation, depression, age and ejection fraction were standardized in the multiple models to compare the importance of the individual covariates on the same scale. After standardization, motivation had a just as high association with $VO_{2peak}$ as depression. Comorbidity also had a high impact on $VO_{2peak}$.

## Change in $VO_{2peak}$ following CR

The mean improvement from CR was 2.27 ml/kg/min (SD±4.3), the equivalent of a 12% increase. In age and sex adjusted analyses, change in $VO_{2peak}$ was positively associated with motivation score (0.57 ml/kg/min per SD), and negatively associated with diabetes.

In the multiple-adjusted model, motivation continued to be statistically associated with $VO_{2peak}$ (0.78 ml/kg/min per SD), whereas depression was not associated. Current smokers and patients with chronic kidney disease or diabetes also improved less. Higher age, higher baseline $VO_{2peak}$ and having diabetes or kidney disease was negatively associated with change in VO2peak.

## Adherence

16 patients (7%) ended the CR program prematurely. This was not significantly associated with level of motivation, but statistical power was limited.

## Discussion

We aimed to investigate whether motivation, measured using BREQ-2, was a predictor of $VO_{2peak}$ before and after CR in an elderly cardiac population. This is the first study to apply BREQ-2 to cardiac patients to predict the success of CR. Using the computed RAI score, we found a significant association between motivation and $VO_{2peak}$, both before and after CR. This was persistent in both simple and multiple-adjusted regression analyses.

## Motivation as a predictor of $VO_{2peak}$ before and after CR

Applying the BREQ-2 to measure motivation seems to be a valid tool to predict physical capacity in the elderly cardiac population and may be a useful assessment tool to target patients with lower motivation who could need special attention during exercise-based rehabilitation.

Psychological distress, measured using depression, anxiety, and vital exhaustion, does not appear to affect the inverse relationship between motivation and $VO_{2peak}$.

In current CR programs, it is recommended that the patients are screened for psychological distress with, for example, the PHQ9 score or the Hospital Anxiety and Depression Scale (HADS) [30]. These results showed that depression is associated with r $VO_{2peak}$ before CR. However, the presence of depression did not significantly affect the impact of motivation on $VO_{2peak}$. Vital exhaustion and anxiety did not have an impact on $VO_{2peak}$ before CR.

Motivation was the only psychological factor that had an impact on change in $VO_{2peak}$ following CR. Neither depression, anxiety, nor vital exhaustion had an impact on change in $VO_{2peak}$. This could suggest that it is more relevant to screen older cardiac patients for motivation rather than other psychological factors, at least if the purpose is to screen for barriers to CR. Our results suggest that the application of BREQ-2 before CR could help the health care

professionals in capturing unmotivated patients and help them frame a rehabilitation that supports the patients in building motivation.

## Constructs of motivation

BREQ-2 measured five constructs of motivation. In addition, we calculated the summed score, RAI. Some literature suggests that applying a simple score, e.g., the RAI, is a step backward and a simplification of the SDT.

We tested all possible constructs of motivation against $VO_{2peak}$ in this paper and found that the constructed score was the only score that was normally distributed, and that this score also had the highest correlation with $VO_{2peak}$ (Fig 1).

The skewness of amotivation, external regulation and intrinsic regulation, in particular, might be explained by the selection of patients. The patients participated voluntarily in the CR program, and this suggests that the patients had at least some motivation for exercise. Patients with a high level of amotivation may be prone to reject participation in CR and participation in the study, introducing an increased risk of selection bias. This suggests that not all constructs of motivation fit equally well for patients engaging in exercise-based CR.

For future research, the model fit for amotivation specifically may be better if the BREQ-2 questionnaire is collected while the patients are still hospitalized. Many cardiac patients never initiate CR, and these patients, in particular, could prove to have higher levels of amotivation.

## Strengths and weaknesses

The focus on adapting a simple screening tool for motivation is a new approach in cardiac rehabilitation. Motivational interviews can be time-consuming, and demand extra resources in a CR unit. Using the BREQ-2 could prove a relevant tool to guide therapists in targeting patients with low levels of motivation before they initiate rehabilitation.

The prospective study design provided the high quality of the data, both with regards to exposures, confounders, and endpoints. Also, given the nature of the prospective design, we could address the issue of causality between exposure and outcome.

Less than 50% of cardiac patients participate in CR [31], a lower proportion of women participated and patients with insufficient understanding of the Danish language were also excluded. While this does not affect internal validity, it may affect generalizability of the results. It is also uncertain whether findings can be transferred to younger cardiac patients (<65 years).

As we investigated an elderly population age-related comorbidities could have an impact on both motivation and outcome. In this study no geriatric assessment was performed on the patients. However, the patients referred to rehabilitation did have an examination with a cardiologist who assessed mental capability to complete the CR program before referral and patients with physical disabilities rendering exercise training impossible were not included.

## Conclusion

Motivation predicted $VO_{2peak}$, both before and after participating in CR. Motivation measured with the BREQ-2 questionnaire may be of value as a screening tool for elderly cardiac patients to identify patients in need of specific attention for successful CR. Future studies should address whether interventions targeting motivation may improve outcomes of CR.

## Supporting information

**S1 Flowchart. Patient population with exclusion and reason for exclusion.**
(TIF)

**S1 Appendix. Behavioral regulation in exercise Questionnaire-2 with 19 questions.** (ZIP)

## Author Contributions

**Conceptualization:** Nicolai Mikkelsen, Christian Have Dall, Marianne Frederiksen, Hanne Rasmusen, Eva Prescott.

**Data curation:** Nicolai Mikkelsen, Marianne Frederiksen.

**Formal analysis:** Nicolai Mikkelsen, Eva Prescott.

**Funding acquisition:** Nicolai Mikkelsen.

**Investigation:** Nicolai Mikkelsen, Christian Have Dall, Marianne Frederiksen, Annette Holdgaard, Hanne Rasmusen, Eva Prescott.

**Methodology:** Nicolai Mikkelsen, Hanne Rasmusen, Eva Prescott.

**Project administration:** Nicolai Mikkelsen, Annette Holdgaard, Hanne Rasmusen, Eva Prescott.

**Resources:** Nicolai Mikkelsen.

**Software:** Nicolai Mikkelsen.

**Supervision:** Christian Have Dall, Eva Prescott.

**Validation:** Nicolai Mikkelsen, Annette Holdgaard.

**Visualization:** Nicolai Mikkelsen.

**Writing – original draft:** Nicolai Mikkelsen, Hanne Rasmusen, Eva Prescott.

**Writing – review & editing:** Nicolai Mikkelsen, Christian Have Dall, Marianne Frederiksen, Annette Holdgaard, Hanne Rasmusen, Eva Prescott.

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
