## [Decision Letter · Decision Letter 0]

18 Mar 2021

PONE-D-21-02329

The motivation for physical activity is a predictor of VO2peak and is a useful parameter when determining the need for cardiac rehabilitation in an elderly cardiac population

PLOS ONE

Dear Dr. Mikkelsen,

Thank you for submitting your manuscript to PLOS ONE. After careful consideration, we feel that it has merit but does not fully meet PLOS ONE’s publication criteria as it currently stands. Therefore, we invite you to submit a revised version of the manuscript that addresses the points raised during the review process.

We look forward to receiving your revised manuscript.

Kind regards,

Delphine De Smedt

Academic Editor

PLOS ONE

Journal Requirements:

We note that you have indicated that data from this study are available upon request. PLOS only allows data to be available upon request if there are legal or ethical restrictions on sharing data publicly. For information on unacceptable data access restrictions, please see http://journals.plos.org/plosone/s/data-availability#loc-unacceptable-data-access-restrictions.

2a) If there are ethical or legal restrictions on sharing a de-identified data set, please explain them in detail (e.g., data contain potentially identifying or sensitive patient information) and who has imposed them (e.g., an ethics committee). Please also provide contact information for a data access committee, ethics committee, or other institutional body to which data requests may be sent.

2b) If there are no restrictions, please upload the minimal anonymized data set necessary to replicate your study findings as either Supporting Information files or to a stable, public repository and provide us with the relevant URLs, DOIs, or accession numbers. Please see http://www.bmj.com/content/340/bmj.c181.long for guidelines on how to de-identify and prepare clinical data for publication. For a list of acceptable repositories, please see http://journals.plos.org/plosone/s/data-availability#loc-recommended-repositories.

Reviewers' comments:

Reviewer's Responses to Questions

**Comments to the Author**

1. Is the manuscript technically sound, and do the data support the conclusions?

Reviewer #1: Partly

Reviewer #2: Yes

2. Has the statistical analysis been performed appropriately and rigorously? 

Reviewer #1: Yes

Reviewer #2: I Don't Know

3. Have the authors made all data underlying the findings in their manuscript fully available?

Reviewer #1: Yes

Reviewer #2: Yes

4. Is the manuscript presented in an intelligible fashion and written in standard English?

Reviewer #1: Yes

Reviewer #2: Yes

5. Review Comments to the Author

Reviewer #1: The authors aimed to investigate the association between motivation, as measured by means of BREQ-2 questionnaire, and (changes in) exercise capacity in older patients participating in an ambulatory phase II cardiac rehabilitation program. A cohort of 203 patients was enrolled of which 182 participants completed the rehabilitation program and follow-up exercise testing. In general, the manuscript is well-written and reads fluently. The following suggestions/remarks are aimed at further improving the clarity of the manuscript.

In the introduction, authors focus mainly on the fact that uptake and adherence to CR is low and that some physiological factors have been identified (e.g. anxiety, depression) that party explain this poor adherence/compliance. Then authors suggest that another, tough less investigated, parameter could be motivation. In the introduction, i miss a bit the rationale why you then opt to investigate the association between motivation and change in VO2/baseline VO2 and why for instance not focusing on the association between motivation and adherence/compliance to CR. Could authors elaborate a bit more on why they opted (rationale) for exercise capacity.

Methods:

Study population. One of the biggest challenges with regard to CR remains uptake of CR in which motivation most likely plays a crucial role. Could authors add how many of the hospitalized patients following ACS,PCT, CABG or valve replacement were in fact eligible for participation in a CR program. It would be particularly interesting to know whether motivation played a crucial role in the uptake of CR. Could it be be that patients already deciding to go to CR are already the more motivated patients. Could there be some selection bias?

Was this the first enrollment in CR program for all patients? If not, please add to table 1 number of patients that participated for the first time and number of patients for whom this was 2nd or 3rd enrollment.

Rehabilitation program. Please add the number of sessions that make up a full rehabilitation program

Primary exposure. Motivation is categorized into three categories: low, medium, high. Was this an arbitrary categorization or according to previous reference. In case of the latter, please add reference.

Please provide a definition for premature ending of the rehab program and a definition for compliance (i.e. how was it calculated). Did you observe a difference in number of patients that prematurely ended the rehab program and with regard to compliance across the three categories of motivation?

Table 1: please add a column with the overall data of the study population. Further, please add a definition for hypertension / hypercholesterolemia. Is there a rationale for only reporting data on statins and beta-blockers but not insulin / other BP lowering medication/... Add SD to ejection fraction. Please also add to this table VO2 peak expressed as percentage of predicted, RER and BORG score.

Study outcome: As peak VO2 is your primary outcome somewhat more detailed information could be provided. E.g each test aimed at RER> 1.1. Was this achieved in all patients? Which protocol did you apply? Next to RER, did you use other parameters to decide on the maximal character of the test? How did you define peakVO2?

Confounding variables. Authors report that physical activity was assessed during leisure time. How was this assessed? Further, was PA in leisure time different across the three categories of motivation? Provide also summary data for PA. Please also add where you retrieved data on demographics, EF etc.

Questionnaires. How where questionnaires completed (online survey, interview, paper,...). Especially in case of paper and pencil - was there any missing data?

Rehabilitation program. Did you monitor your patients during the training session? If so, please provide details on what was monitored during training and whether patients were adherent to the prescribed intensity/duration/...of the exercise program

Please add the statistical tests used to evaluate normality of your data. Which data were not normally distributed?

It seems that Figure 1 is missing?

Though i agree and understand that you should not repeating what is in the tables, yet somewhat more statistics and numerical data in the results section could improve the reading without having to go back to the table all the time.

In the discussion section, the different subsections of of motivation are discussed - though this is missing in the results section. please provide the results first before discussing it.

Reviewer #2: This paper by Mikkelsen et al prospectively investigated the correlation of individual motivation for physical activity, determined as the summed RAI score by BREQ-2 questionnaire, and the cardio-pulmonary fitness, measured as VO2peak in a symptom-limited CPET, in 203 patients participating in an ambulatory cardiac rehabilitation (CR) program. They found that in contrast to other psychological determinants such as depression and anxiety, motivation was and independent predictor of VO2peak at baseline and predicted delta VO2peak from baseline to the end of CR. The authors conclude that motivation measured with the BREQ-2 questionnaire can be useful as a screening tool for

elderly cardiac patients undergoing CR to detect patients with lower potential to increase VO2peak.

The study addresses an important topic and provides interesting and clinically useful new data to the field of CR. The prospective study design is solid. The study, however, includes a very selected patient population (e.g. > 64 years, only 9 % current smokers, high level of education) that might not be representative for most CR cohorts.

Major points

The main limitation of the study is its low patient number and its selective recruitment. Considering the study duration of 26 months, the inclusion of 203 patients seems rather low (average of < 8 patients/months). What is the size of the local CR program? Where patients recruited consecutively every day or dependent on the availability of the study staff?

A better motivation could result in the achievement of a higher respiratory exchange ratio (RER) in CPET. Did RAI correlate with RER. This relation should be presented in the data.

Please specify the “multiple adjustment” made in the multiple adjusted model. Was the model also adjusted for diabetes, body mass index and activity level? If not, this should be done.

Was there a signal of lower RAI in the 16 drop outs?

Did the CR program had an impact on the RAI score? Providing data on RAI scores at baseline and after CR would be interesting

Minor points

Why were patients < 64 years excluded? What is the percentage of patients < 64 years in the CR program

Appendix is missing

Figure 1 should be presented as a flow chart

The low numbers of female participants should be stated as a limitation

6. PLOS authors have the option to publish the peer review history of their article (what does this mean?). If published, this will include your full peer review and any attached files.

Reviewer #1: No

Reviewer #2: No

---

## [Author Response · Author response to Decision Letter 0]

19 Aug 2021

Response to reviewers

Reviewer #1: 

The authors aimed to investigate the association between motivation, as measured by means of BREQ-2 questionnaire, and (changes in) exercise capacity in older patients participating in an ambulatory phase II cardiac rehabilitation program. A cohort of 203 patients was enrolled of which 182 participants completed the rehabilitation program and follow-up exercise testing. In general, the manuscript is well-written and reads fluently. The following suggestions/remarks are aimed at further improving the clarity of the manuscript.

• Comment 1: In the introduction, authors focus mainly on the fact that uptake and adherence to CR is low and that some physiological factors have been identified (e.g. anxiety, depression) that party explain this poor adherence/compliance. Then authors suggest that another, tough less investigated, parameter could be motivation. In the introduction, i miss a bit the rationale why you then opt to investigate the association between motivation and change in VO2/baseline VO2 and why for instance not focusing on the association between motivation and adherence/compliance to CR. Could authors elaborate a bit more on why they opted (rationale) for exercise capacity.

• Response: Thank you for this relevant comment. We have addressed this in the introduction now with the rationale that VO2peak/change in VO2peak is a great measure for the effect of a CR program and the fact that VO2peak is an important predictor of future morbidity and mortality.

• Comment 2: Study population. One of the biggest challenges with regard to CR remains uptake of CR in which motivation most likely plays a crucial role. Could authors add how many of the hospitalized patients following ACS,PCT, CABG or valve replacement were in fact eligible for participation in a CR program. It would be particularly interesting to know whether motivation played a crucial role in the uptake of CR. Could it be be that patients already deciding to go to CR are already the more motivated patients. Could there be some selection bias?

• Response: Unfortunately, this cannot be addressed in this paper. We very much agree that there can be selection bias on attendance in CR because we capture the most motivated.

We only had the opportunity to address motivation among patients that attended CR. If possible, it would be very interesting to interview patients with the BREQ-2 questionnaire just after surgery or even before if possible, to see if there was a correlation between motivation for exercise and attendance in CR. This could be an opportunity to capture patients with a lack of motivation and perform some kind of intervention to increase motivation.

• Comment 3: Was this the first enrolment in CR program for all patients? If not, please add to table 1 number of patients that participated for the first time and number of patients for whom this was 2nd or 3rd enrolment.

• Response: We do not have this information available. However, the normal clinical practice at this hospital is that if patients are referred for the second or third time, they will be referred for municipal rehabilitation and not at the hospital. Thus, for the majority of the patients this will be their first CR. We have added this information to the methods section: Study population

• Comment 4: Rehabilitation program. Please add the number of sessions that make up a full rehabilitation program

• Response: Thank you for your comment as this is indeed relevant. In the description of the rehabilitation program we do state that the program consists of an 8-week intervention with 2 weekly sessions of 1.5 hours. We have added the total of 16 sessions to the text as well. I hope this is what you alluded to. 

• Comment 5: Primary exposure. Motivation is categorized into three categories: low, medium, high. Was this an arbitrary categorization or according to previous reference. In case of the latter, please add reference.

• Response: The BREQ-2 questionnaire has not previously been divided into groups. The three groups were created by the authors in order to provide a descriptive overview of the population. This should of course be stated in the article and we have now done this in the methods section: Primary exposure – the motivation for physical activity

• Question 6: Please provide a definition for premature ending of the rehab program and a definition for compliance (i.e. how was it calculated). Did you observe a difference in number of patients that prematurely ended the rehab program and with regard to compliance across the three categories of motivation?

• Response: Compliance is described in confounding variables and was measured as attendance (<50%, 50-75% or >75% attendance). We have added information on subjects that prematurely ended the program to the methods section: Rehabilitation program. The categories of motivation were not significantly associated with compliance.

• Comment 7: Table 1: please add a column with the overall data of the study population. Further, please add a definition for hypertension / hypercholesterolemia. Is there a rationale for only reporting data on statins and beta-blockers but not insulin / other BP lowering medication/... Add SD to ejection fraction. Please also add to this table VO2 peak expressed as percentage of predicted, RER and BORG score.

• Response: We have added a total study population column and added information on hypertension / hypercholesterolemia and predicted VO2peak, RER and BORG score. We reported data on statins and beta-blockers because these are guidelines recommended medications for (almost) all patients with CVD and frequently used as indicators. 

• Comment 8: Study outcome: As peak VO2 is your primary outcome somewhat more detailed information could be provided. E.g each test aimed at RER> 1.1. Was this achieved in all patients? Which protocol did you apply? Next to RER, did you use other parameters to decide on the maximal character of the test? How did you define peakVO2?

• Response: We have added this information in the methods section: Study outcome

• Comment 9: Confounding variables. Authors report that physical activity was assessed during leisure time. How was this assessed? Further, was PA in leisure time different across the three categories of motivation? Provide also summary data for PA. Please also add where you retrieved data on demographics, EF etc.

• Response: This valid point has been addressed in table 1. There was indeed a positive trend correlating difference between motivation for physical activity and self-reported physical activity, although this was not significant.

• Comment 10: Questionnaires. How where questionnaires completed (online survey, interview, paper,...). Especially in case of paper and pencil - was there any missing data?

• Response: The questionnaires were primarily filled out online. One a few subjects filled out the questionnaire in a paper format as they did not have access to a computer. We have described the missing data in figure 1 as exclusions from the analyses. This comprised of 13 patients that did not answer. 

• Comment 11: Rehabilitation program. Did you monitor your patients during the training session? If so, please provide details on what was monitored during training and whether patients were adherent to the prescribed intensity/duration/...of the exercise program

• Response: The patients were not monitored during the training sessions. The therapists supervised all training sessions and guided the patients based on the BORG scale. We have added this information to the methods section: rehabilitation program

• Comment 12: Please add the statistical tests used to evaluate normality of your data. Which data were not normally distributed?

• Response: I believe we have described this in under: Statistical analyses “Normally distributed variables were compared across the different levels of motivation using one-way ANOVA. Non-normally distributed variables were tested using Mann-Whitney and Kruskal-Wallis tests, while a Chi 2 test tested categorical data. Statistically significant differences between the groups were tested with pairwise comparisons using t test and Chi 2 tests.”

• Comment 13: It seems that Figure 1 is missing?

• Response: We apologize if this was not visible to you. It was in the compiled sheet approved before uploading. We will contact the editor to ensure that all figures are visible in the revised paper.

• Comment 14: Though I agree and understand that you should not repeating what is in the tables, yet somewhat more statistics and numerical data in the results section could improve the reading without having to go back to the table all the time.

• Response: We have tried to accommodate this by adding more numbers to the results section.

• Comment 15: In the discussion section, the different subsections of of motivation are discussed - though this is missing in the results section. please provide the results first before discussing it.

• Response: I do believe that we have described this in the results section: Constructs of motivation. Unless I am misunderstanding your comment

Reviewer #2: 

This paper by Mikkelsen et al prospectively investigated the correlation of individual motivation for physical activity, determined as the summed RAI score by BREQ-2 questionnaire, and the cardio-pulmonary fitness, measured as VO2peak in a symptom-limited CPET, in 203 patients participating in an ambulatory cardiac rehabilitation (CR) program. They found that in contrast to other psychological determinants such as depression and anxiety, motivation was and independent predictor of VO2peak at baseline and predicted delta VO2peak from baseline to the end of CR. The authors conclude that motivation measured with the BREQ-2 questionnaire can be useful as a screening tool for

elderly cardiac patients undergoing CR to detect patients with lower potential to increase VO2peak.

The study addresses an important topic and provides interesting and clinically useful new data to the field of CR. The prospective study design is solid. The study, however, includes a very selected patient population (e.g. > 64 years, only 9 % current smokers, high level of education) that might not be representative for most CR cohorts.

Major points

• Comment 1: The main limitation of the study is its low patient number and its selective recruitment. Considering the study duration of 26 months, the inclusion of 203 patients seems rather low (average of < 8 patients/months). What is the size of the local CR program? Where patients recruited consecutively every day or dependent on the availability of the study staff?

• Response: This was a selective recruitment because this is a sub-study of the EU-CaRE trial that aimed for including patients >64 years which only is a little more than half of the patients being offered CR (. 40 % of the population is <64 years). Patients were recruited consecutively by the cardiologist and if the patient was eligible and wished to participate in CR they were asked to participate. Almost all CR patients agreed to participate. We did also publish this in an article with baseline data from the EU-CaRE cohort:

Prescott E, Mikkelsen N, Holdgaard A, Eser P, Marcin T, Wilhelm M, Gil CP, González-Juanatey JR, Moatemri F, Iliou MC, Schneider S, Schromm E, Zeymer U, Meindersma EP, Ardissino D, Kolkman EK, Prins LF, van der Velde AE, Van 't Hof AW, de Kluiver EP. Cardiac rehabilitation in the elderly patient in eight rehabilitation units in Western Europe: Baseline data from the EU-CaRE multicentre observational study. Eur J Prev Cardiol. 2019 Jul;26(10):1052-1063. doi: 10.1177/2047487319839819. Epub 2019 Mar 29. PMID: 30924688.

• Comment 2: A better motivation could result in the achievement of a higher respiratory exchange ratio (RER) in CPET. Did RAI correlate with RER. This relation should be presented in the data.

• Response: This was added to table 1. There was no significant difference between groups.

• Comment 3: Please specify the “multiple adjustment” made in the multiple adjusted model. Was the model also adjusted for diabetes, body mass index and activity level? If not, this should be done.

• Response: All relevant confounding variables describes in table 1 were tested in the multiple models to see if it made any significant change to the effect motivation had on peak VO2. We have added the confounding variables of interest in the statistical analyses section. The first model for baseline VO2peak was both adjusted for diabetes and self-reported physical activity at these had significant confounding impact. BMI was also tested, but not included as it did not impact the result. The first model for baseline VO2peak was both adjusted for diabetes and self-reported physical activity at these had significant confounding impact. BMI was also tested, but not included as it did not impact the result.

• Comment 4: Was there a signal of lower RAI in the 16 dropouts?

• Response: Thank you for this comment. There was an approximately even distribution among dropouts irrespectively of level of motivation. COPD and diabetes were associated with ending prematurely. This was not significant, probably due to low power of this analyses. We have added this to our results section 

• Comment 5: Did the CR program had an impact on the RAI score? Providing data on RAI scores at baseline and after CR would be interesting

• Response: This is a very interesting comment. We only have data on the baseline motivation. But this bring a new interesting aspect to rehabilitation and motivation since motivation also can change over time when exposed to changes in life. 

Minor points

• Comment 6: Why were patients < 64 years excluded? What is the percentage of patients < 64 years in the CR program

• Response: This is because this a Danish sub-study based on the EU-CaRE study which is a multi-country study on elderly cardiac patient >64 years of age. In this hospital rehabilitation app. 40 % of the population is <64 years. 

• Comment 7: Appendix is missing

• Response: I am sorry for that. This must have been missed in the upload process. I will make sure it becomes available to you.

• Comment 8: Figure 1 should be presented as a flow chart

• Response: Valid point and this has been corrected.

• Comment 9: The low numbers of female participants should be stated as a limitation

• Response: Yes. This is a general limitation for rehabilitation, and we have added this to limitations.

---

## [Editor Report · Decision Letter 1]

21 Dec 2021

PONE-D-21-02329R1

The motivation for physical activity is a predictor of VO2peak and is a useful parameter when determining the need for cardiac rehabilitation in an elderly cardiac population

PLOS ONE

Dear Dr.Mikkelsen

Thank you for submitting your manuscript to PLOS ONE. After careful consideration, we have decided that your manuscript does not meet our criteria for publication and must therefore be rejected.

Specifically:

I am sorry that we cannot be more positive on this occasion, but hope that you appreciate the reasons for this decision.

Yours sincerely,

Xianwu Cheng, M.D., Ph.D., FAHA

Academic Editor

PLOS ONE

Additional Editor Comments (if provided):

Both original reviewers have been declined to review this revised manuscript. The main serious problems are that the authors changed key data without acceptable explanation (especially, in Table).

- - - - -

---

## [Author Response · Author response to Decision Letter 1]

21 Feb 2022

Response to reviewers

Reviewer #1: 

The authors aimed to investigate the association between motivation, as measured by means of BREQ-2 questionnaire, and (changes in) exercise capacity in older patients participating in an ambulatory phase II cardiac rehabilitation program. A cohort of 203 patients was enrolled of which 182 participants completed the rehabilitation program and follow-up exercise testing. In general, the manuscript is well-written and reads fluently. The following suggestions/remarks are aimed at further improving the clarity of the manuscript.

• Comment 1: In the introduction, authors focus mainly on the fact that uptake and adherence to CR is low and that some physiological factors have been identified (e.g. anxiety, depression) that party explain this poor adherence/compliance. Then authors suggest that another, tough less investigated, parameter could be motivation. In the introduction, i miss a bit the rationale why you then opt to investigate the association between motivation and change in VO2/baseline VO2 and why for instance not focusing on the association between motivation and adherence/compliance to CR. Could authors elaborate a bit more on why they opted (rationale) for exercise capacity.

• Response: Thank you for this relevant comment. We have addressed this in the introduction now with the rationale that VO2peak/change in VO2peak is a great measure for the effect of a CR program and the fact that VO2peak is an important predictor of future morbidity and mortality.

• Comment 2: Study population. One of the biggest challenges with regard to CR remains uptake of CR in which motivation most likely plays a crucial role. Could authors add how many of the hospitalized patients following ACS,PCT, CABG or valve replacement were in fact eligible for participation in a CR program. It would be particularly interesting to know whether motivation played a crucial role in the uptake of CR. Could it be be that patients already deciding to go to CR are already the more motivated patients. Could there be some selection bias?

• Response: Unfortunately, this cannot be addressed in this paper. We very much agree that there can be selection bias on attendance in CR because we capture the most motivated.

We only had the opportunity to address motivation among patients that attended CR. If possible, it would be very interesting to interview patients with the BREQ-2 questionnaire just after surgery or even before if possible, to see if there was a correlation between motivation for exercise and attendance in CR. This could be an opportunity to capture patients with a lack of motivation and perform some kind of intervention to increase motivation.

• Comment 3: Was this the first enrolment in CR program for all patients? If not, please add to table 1 number of patients that participated for the first time and number of patients for whom this was 2nd or 3rd enrolment.

• Response: We do not have this information available. However, the normal clinical practice at this hospital is that if patients are referred for the second or third time, they will be referred for municipal rehabilitation and not at the hospital. Thus, for the majority of the patients this will be their first CR. We have added this information to the methods section: Study population

• Comment 4: Rehabilitation program. Please add the number of sessions that make up a full rehabilitation program

• Response: Thank you for your comment as this is indeed relevant. In the description of the rehabilitation program we do state that the program consists of an 8-week intervention with 2 weekly sessions of 1.5 hours. We have added the total of 16 sessions to the text as well. I hope this is what you alluded to. 

• Comment 5: Primary exposure. Motivation is categorized into three categories: low, medium, high. Was this an arbitrary categorization or according to previous reference. In case of the latter, please add reference.

• Response: The BREQ-2 questionnaire has not previously been divided into groups. The three groups were created by the authors in order to provide a descriptive overview of the population. This should of course be stated in the article and we have now done this in the methods section: Primary exposure – the motivation for physical activity

• Question 6: Please provide a definition for premature ending of the rehab program and a definition for compliance (i.e. how was it calculated). Did you observe a difference in number of patients that prematurely ended the rehab program and with regard to compliance across the three categories of motivation?

• Response: Compliance is described in confounding variables and was measured as attendance (<50%, 50-75% or >75% attendance). We have added information on subjects that prematurely ended the program to the methods section: Rehabilitation program. The categories of motivation were not significantly associated with compliance.

• Comment 7: Table 1: please add a column with the overall data of the study population. Further, please add a definition for hypertension / hypercholesterolemia. Is there a rationale for only reporting data on statins and beta-blockers but not insulin / other BP lowering medication/... Add SD to ejection fraction. Please also add to this table VO2 peak expressed as percentage of predicted, RER and BORG score.

• Response: We have added a total study population column and added information on hypertension / hypercholesterolemia and predicted VO2peak, RER and BORG score. We reported data on statins and beta-blockers because these are guidelines recommended medications for (almost) all patients with CVD and frequently used as indicators. 

• Comment 8: Study outcome: As peak VO2 is your primary outcome somewhat more detailed information could be provided. E.g each test aimed at RER> 1.1. Was this achieved in all patients? Which protocol did you apply? Next to RER, did you use other parameters to decide on the maximal character of the test? How did you define peakVO2?

• Response: We have added this information in the methods section: Study outcome

• Comment 9: Confounding variables. Authors report that physical activity was assessed during leisure time. How was this assessed? Further, was PA in leisure time different across the three categories of motivation? Provide also summary data for PA. Please also add where you retrieved data on demographics, EF etc.

• Response: This valid point has been addressed in table 1. There was indeed a positive trend correlating difference between motivation for physical activity and self-reported physical activity, although this was not significant.

• Comment 10: Questionnaires. How where questionnaires completed (online survey, interview, paper,...). Especially in case of paper and pencil - was there any missing data?

• Response: The questionnaires were primarily filled out online. One a few subjects filled out the questionnaire in a paper format as they did not have access to a computer. We have described the missing data in figure 1 as exclusions from the analyses. This comprised of 13 patients that did not answer. 

• Comment 11: Rehabilitation program. Did you monitor your patients during the training session? If so, please provide details on what was monitored during training and whether patients were adherent to the prescribed intensity/duration/...of the exercise program

• Response: The patients were not monitored during the training sessions. The therapists supervised all training sessions and guided the patients based on the BORG scale. We have added this information to the methods section: rehabilitation program

• Comment 12: Please add the statistical tests used to evaluate normality of your data. Which data were not normally distributed?

• Response: I believe we have described this in under: Statistical analyses “Normally distributed variables were compared across the different levels of motivation using one-way ANOVA. Non-normally distributed variables were tested using Mann-Whitney and Kruskal-Wallis tests, while a Chi 2 test tested categorical data. Statistically significant differences between the groups were tested with pairwise comparisons using t test and Chi 2 tests.”

• Comment 13: It seems that Figure 1 is missing?

• Response: We apologize if this was not visible to you. It was in the compiled sheet approved before uploading. We will contact the editor to ensure that all figures are visible in the revised paper.

• Comment 14: Though I agree and understand that you should not repeating what is in the tables, yet somewhat more statistics and numerical data in the results section could improve the reading without having to go back to the table all the time.

• Response: We have tried to accommodate this by adding more numbers to the results section.

• Comment 15: In the discussion section, the different subsections of of motivation are discussed - though this is missing in the results section. please provide the results first before discussing it.

• Response: I do believe that we have described this in the results section: Constructs of motivation. Unless I am misunderstanding your comment

Reviewer #2: 

This paper by Mikkelsen et al prospectively investigated the correlation of individual motivation for physical activity, determined as the summed RAI score by BREQ-2 questionnaire, and the cardio-pulmonary fitness, measured as VO2peak in a symptom-limited CPET, in 203 patients participating in an ambulatory cardiac rehabilitation (CR) program. They found that in contrast to other psychological determinants such as depression and anxiety, motivation was and independent predictor of VO2peak at baseline and predicted delta VO2peak from baseline to the end of CR. The authors conclude that motivation measured with the BREQ-2 questionnaire can be useful as a screening tool for

elderly cardiac patients undergoing CR to detect patients with lower potential to increase VO2peak.

The study addresses an important topic and provides interesting and clinically useful new data to the field of CR. The prospective study design is solid. The study, however, includes a very selected patient population (e.g. > 64 years, only 9 % current smokers, high level of education) that might not be representative for most CR cohorts.

Major points

• Comment 1: The main limitation of the study is its low patient number and its selective recruitment. Considering the study duration of 26 months, the inclusion of 203 patients seems rather low (average of < 8 patients/months). What is the size of the local CR program? Where patients recruited consecutively every day or dependent on the availability of the study staff?

• Response: This was a selective recruitment because this is a sub-study of the EU-CaRE trial that aimed for including patients >64 years which only is a little more than half of the patients being offered CR (. 40 % of the population is <64 years). Patients were recruited consecutively by the cardiologist and if the patient was eligible and wished to participate in CR they were asked to participate. Almost all CR patients agreed to participate. We did also publish this in an article with baseline data from the EU-CaRE cohort:

Prescott E, Mikkelsen N, Holdgaard A, Eser P, Marcin T, Wilhelm M, Gil CP, González-Juanatey JR, Moatemri F, Iliou MC, Schneider S, Schromm E, Zeymer U, Meindersma EP, Ardissino D, Kolkman EK, Prins LF, van der Velde AE, Van 't Hof AW, de Kluiver EP. Cardiac rehabilitation in the elderly patient in eight rehabilitation units in Western Europe: Baseline data from the EU-CaRE multicentre observational study. Eur J Prev Cardiol. 2019 Jul;26(10):1052-1063. doi: 10.1177/2047487319839819. Epub 2019 Mar 29. PMID: 30924688.

• Comment 2: A better motivation could result in the achievement of a higher respiratory exchange ratio (RER) in CPET. Did RAI correlate with RER. This relation should be presented in the data.

• Response: This was added to table 1. There was no significant difference between groups.

• Comment 3: Please specify the “multiple adjustment” made in the multiple adjusted model. Was the model also adjusted for diabetes, body mass index and activity level? If not, this should be done.

• Response: All relevant confounding variables describes in table 1 were tested in the multiple models to see if it made any significant change to the effect motivation had on peak VO2. We have added the confounding variables of interest in the statistical analyses section. The first model for baseline VO2peak was both adjusted for diabetes and self-reported physical activity at these had significant confounding impact. BMI was also tested, but not included as it did not impact the result. The first model for baseline VO2peak was both adjusted for diabetes and self-reported physical activity at these had significant confounding impact. BMI was also tested, but not included as it did not impact the result.

• Comment 4: Was there a signal of lower RAI in the 16 dropouts?

• Response: Thank you for this comment. There was an approximately even distribution among dropouts irrespectively of level of motivation. COPD and diabetes were associated with ending prematurely. This was not significant, probably due to low power of this analyses. We have added this to our results section 

• Comment 5: Did the CR program had an impact on the RAI score? Providing data on RAI scores at baseline and after CR would be interesting

• Response: This is a very interesting comment. We only have data on the baseline motivation. But this bring a new interesting aspect to rehabilitation and motivation since motivation also can change over time when exposed to changes in life. 

Minor points

• Comment 6: Why were patients < 64 years excluded? What is the percentage of patients < 64 years in the CR program

• Response: This is because this a Danish sub-study based on the EU-CaRE study which is a multi-country study on elderly cardiac patient >64 years of age. In this hospital rehabilitation app. 40 % of the population is <64 years. 

• Comment 7: Appendix is missing

• Response: I am sorry for that. This must have been missed in the upload process. I will make sure it becomes available to you.

• Comment 8: Figure 1 should be presented as a flow chart

• Response: Valid point and this has been corrected.

• Comment 9: The low numbers of female participants should be stated as a limitation

• Response: Yes. This is a general limitation for rehabilitation, and we have added this to limitations.

---

## [Decision Letter · Decision Letter 2]

25 Jul 2022

PONE-D-21-02329R2The motivation for physical activity is a predictor of VO2peak and is a useful parameter when determining the need for cardiac rehabilitation in an elderly cardiac populationPLOS ONE

Dear Dr. Mikkelsen,

Thank you for submitting your manuscript to PLOS ONE. After careful consideration, we feel that it has merit but does not fully meet PLOS ONE’s publication criteria as it currently stands. Therefore, we invite you to submit a revised version of the manuscript that addresses the points raised during the review process.

Dear authors, Thank you for this revised manuscript. I agree with the reviewers that the comments of the previous reviewers have been sufficiently taken into account. Some minor issues in particular on the characteristics of the population studied still need to be addressed (see 6. Review Comments to the Author). I am sure you will be able to deal with them easily.

We look forward to receiving your revised manuscript.

Kind regards,

Annett Salzwedel

Academic Editor

PLOS ONE

Journal Requirements:

Additional Editor Comments (if provided):

Reviewers' comments:

Reviewer's Responses to Questions

**Comments to the Author**

1. If the authors have adequately addressed your comments raised in a previous round of review and you feel that this manuscript is now acceptable for publication, you may indicate that here to bypass the “Comments to the Author” section, enter your conflict of interest statement in the “Confidential to Editor” section, and submit your "Accept" recommendation.

Reviewer #3: All comments have been addressed

Reviewer #4: (No Response)

2. Is the manuscript technically sound, and do the data support the conclusions?

Reviewer #3: Yes

Reviewer #4: Yes

3. Has the statistical analysis been performed appropriately and rigorously? 

Reviewer #3: Yes

Reviewer #4: Yes

4. Have the authors made all data underlying the findings in their manuscript fully available?

Reviewer #3: Yes

Reviewer #4: Yes

5. Is the manuscript presented in an intelligible fashion and written in standard English?

Reviewer #3: Yes

Reviewer #4: Yes

6. Review Comments to the Author

Reviewer #3: The authors attempt to identify a correlation between patient motivation and aerobic exercise capacity, specifically VO2 peak at baseline and change at the end of the cardiac rehabilitation program. It is well structured, novel, and written in good english. All the concerns addressed by the previous round of reviewers have been addressed adequately. Admittedly, the subgroup used for analysis restricts extrapolating this data to a younger age bracket, unfortunately in a male dominant population. Nevertheless, I do believe that the manuscript is of sufficient quality to warrant publication.

Reviewer #4: Interesting study about a clinically elevant topic. Impaired adherence of CR programs is one of the major issues when planning this kind of health resources. Describing predictors of success is very important. We know that CR adherence is low, even in randomized trials. I suggest to cite

Sanchis J, Sastre C, Ruescas A, Ruiz V, Valero E, Bonanad C, García-Blas S, Fernández-Cisnal A, González J, Miñana G, Núñez J. Randomized Comparison of Exercise Intervention Versus Usual Care in Older Adult Patients with Frailty After Acute Myocardial Infarction. Am J Med. 2021 Mar;134(3):383-390.e2. doi: 10.1016/j.amjmed.2020.09.019.

I have some minor comments to the authors

-Please explain the cutt-of of age. In this sense, 64 years is a very young cut off to be designated as elderly

-When talking about patients at older ages, frailty, comrbidity and otger geriatric syndromes are closely related to prognosis, to the need for CR and probably to the motivation for CR. I understand the authors do not have information about geriatric assessment. In this case, this should be inlcuded as a limitation of this study

-Finally the authos should clearly discuss the clinical implications of their findings. Which is the recommendation for patients with low motivation values?. Probably these are the patients which higher need for CR...

7. PLOS authors have the option to publish the peer review history of their article (what does this mean?). If published, this will include your full peer review and any attached files.

Reviewer #3: **Yes: **Dr Mark Abela MD (Melit) MRCP (London) MSc Internal Medicine (Edinburgh) MSc Sports Cardiology (London)

Reviewer #4: No

---

## [Author Response · Author response to Decision Letter 2]

8 Sep 2022

Comment

Reviewer #4: Interesting study about a clinically elevant topic. Impaired adherence of CR programs is one of the major issues when planning this kind of health resources. Describing predictors of success is very important. We know that CR adherence is low, even in randomized trials. I suggest to cite

Sanchis J, Sastre C, Ruescas A, Ruiz V, Valero E, Bonanad C, García-Blas S, Fernández-Cisnal A, González J, Miñana G, Núñez J. Randomized Comparison of Exercise Intervention Versus Usual Care in Older Adult Patients with Frailty After Acute Myocardial Infarction. Am J Med. 2021 Mar;134(3):383-390.e2. doi: 10.1016/j.amjmed.2020.09.019.

Answer: 

Thank you for providing this interesting reference. It is indeed what we see as well in our clinic and we have sited Sanchis et al in the “Introduction” section on page 3.

I have some minor comments to the authors

Comment:

-Please explain the cutt-of of age. In this sense, 64 years is a very young cut off to be designated as elderly

Answer: 

Thank you for raising this question. We agree that in many patients a cut-off of 65 is relatively low, however, there is considerable biological variation. Age 65 is the age of retirement in many European countries. Most importantly, however, is that the cut-of age of more than 64 years was decided in the EU-CaRE study group, which this paper derived data from. As EU-CaRE was a multi-country program, the 8 different countries had to agree of an acceptable cut-of age that all countries could accept. The definition of elderly is subject to discussion and we appreciate your relevant comment.

Comment:

-When talking about patients at older ages, frailty, comorbidity and other geriatric syndromes are closely related to prognosis, to the need for CR and probably to the motivation for CR. I understand the authors do not have information about geriatric assessment. In this case, this should be included as a limitation of this study

Answer: Thank you for this comment. We have added this to our limitations as we did not have a direct geriatric assessment. However, all patients did consult a cardiologist before CR who assessed if the patient were capable, also mentally, to complete a CR program.

Comment:

-Finally the authos should clearly discuss the clinical implications of their findings. Which is the recommendation for patients with low motivation values?. Probably these are the patients which higher need for CR...

Answer: 

This is an interesting point and we have elaborated on the implications on page 9 under discussion and “Motivation as a predictor of VO2peak before and after CR”

---

## [Editor Report · Decision Letter 3]

11 Sep 2022

The motivation for physical activity is a predictor of VO2peak and is a useful parameter when determining the need for cardiac rehabilitation in an elderly cardiac population

PONE-D-21-02329R3

Dear Dr. Mikkelsen,

We’re pleased to inform you that your manuscript has been judged scientifically suitable for publication and will be formally accepted for publication once it meets all outstanding technical requirements.

Kind regards,

Annett Salzwedel

Academic Editor

PLOS ONE
---

## [Editor Report · Acceptance letter]

19 Sep 2022

PONE-D-21-02329R3 

The motivation for physical activity is a predictor of VO2peak and is a useful parameter when determining the need for cardiac rehabilitation in an elderly cardiac population 

Dear Dr. Mikkelsen:

I'm pleased to inform you that your manuscript has been deemed suitable for publication in PLOS ONE. Congratulations! Your manuscript is now with our production department. 

Kind regards, 

on behalf of

Dr. Annett Salzwedel 

Academic Editor

PLOS ONE